# *Candidatus* Desulforudis audaxviator dominates a 975 m deep groundwater community in central Sweden
George Westmeijer [1,6] ✉, Femke van Dam [2], Riikka Kietäväinen [3,4], Carolina González-Rosales[1], Stefan Bertilsson [5], Henrik Drake [2] & Mark Dopson [1]

The continental bedrock contains groundwater-bearing fractures that are home to microbial populations that are vital in mediating the Earth's biogeochemical cycles. However, their diversity is poorly understood due to the difficulty of obtaining samples from this environment. Here, a groundwater-bearing fracture at 975 m depth was isolated by employing packers in order to characterize the microbial community via metagenomes combined with prokaryotic and eukaryotic marker genes (16S and 18S ribosomal RNA gene). Genome-resolved analyses revealed a community dominated by sulfate-reducing Bacillota, predominantly represented by *Candidatus* Desulforudis audaxviator and with Wood-Ljungdahl as the most prevalent pathway for inorganic carbon fixation. Moreover, the eukaryotic community had a considerable diversity and was comprised of mainly flatworms, chlorophytes, crustaceans, ochrophytes, and fungi. These findings support the important role of the Bacillota, with the sulfate reducer *Candidatus* Desulforudis audaxviator as its main representative, as primary producers in the often energy-limited groundwaters of the continental subsurface.

Microorganisms play a vital role in the deep subsurface, a largely unexplored biome up to several kilometers beneath the Earth's surface[1]. Despite being predominantly energy and nutrient limited, the deep subsurface hosts specialized populations that are involved in biogeochemical cycling by mediating the transformation of organic and inorganic matter[2]. Focusing on the continental subsurface, recent studies have used former mines[3–6], boreholes[7–11], and dedicated underground laboratories[12–14] to show that deep subsurface communities can be low diversity such that they are dominated by one bacterial population[15] or methanogenic archaea[16], mixed bacterial communities active in sulfur cycling[17], or be predominantly eukaryotic[18]. Depending on the host rock, geogenic hydrogen from the deep subsurface can be the main energy source for deep subsurface life[19] with chemolithotrophs such as sulfate reducers, methanogens, and acetogens competing for hydrogen to fuel reduction of sulfate or carbon dioxide. As geogenic hydrogen production by rock-water interactions is generally limiting[20], groundwater circulation is key as it is thought to support these hydrogen-dependent populations[16]. Apart from certain sedimentary reservoirs, e.g., oil reservoirs[21], organic carbon is usually scarce or of a refractory nature and autotrophic growth tends to be abundant, if not dominant, in the deep subsurface[22,23]. The concentration of molecular oxygen dictates the mechanism of inorganic carbon fixation as, for example, the Wood-Ljungdahl pathway requires strict anoxia and is favored by prokaryotes in energy-limited conditions[24]. Additionally, it has been suggested that anaerobic fungi could produce hydrogen during respiration[25] and in line with this, consortia of fungi with hydrogenotrophic sulfate reducers[26] and methanogens[27] have been described in the continental subsurface.

The Precambrian craton, the Fennoscandian Shield, encompasses 900 to 3100 Ma old terranes in Norway, Sweden, Finland, and Russia. In terms of tectonic activity, the Fennoscandian Shield is among the most stable geological regions on Earth, with little tectonic activity since the formation of the Caledonian orogen in the Silurian[28]. Previous studies on subsurface groundwaters in the Fennoscandian Shield showed a decrease of microbial abundance[29,30] and a changing microbial community[31,32] with depth. For example, a study on groundwater-bearing fractures intersected by the Outokumpu borehole (Finland) revealed cell numbers to decrease with depth and a microbial community closer to the surface (0–300 m depth) dominated by Alpha- and Betaproteobacteria while the Bacillota (synonym Firmicutes) were abundant below 900 m depth[29]. In addition,

[1]Centre for Ecology and Evolution in Microbial model Systems (EEMiS), Linnaeus University, Stuvaregatan 4, Kalmar, Sweden. [2]Department of Biology and Environmental Sciences, Linnaeus University, Stuvaregatan 4, Kalmar, Sweden. [3]Geological Survey of Finland, Espoo, Finland. [4]Department of Geosciences and Geography, University of Helsinki, Helsinki, Finland. [5]Department of Aquatic Sciences and Assessment, Swedish University of Agricultural Sciences, Uppsala, Sweden. [6]Present address: Department of Chemistry, Umeå University, Umeå, Sweden. ✉e-mail: george.westmeijer@umu.se

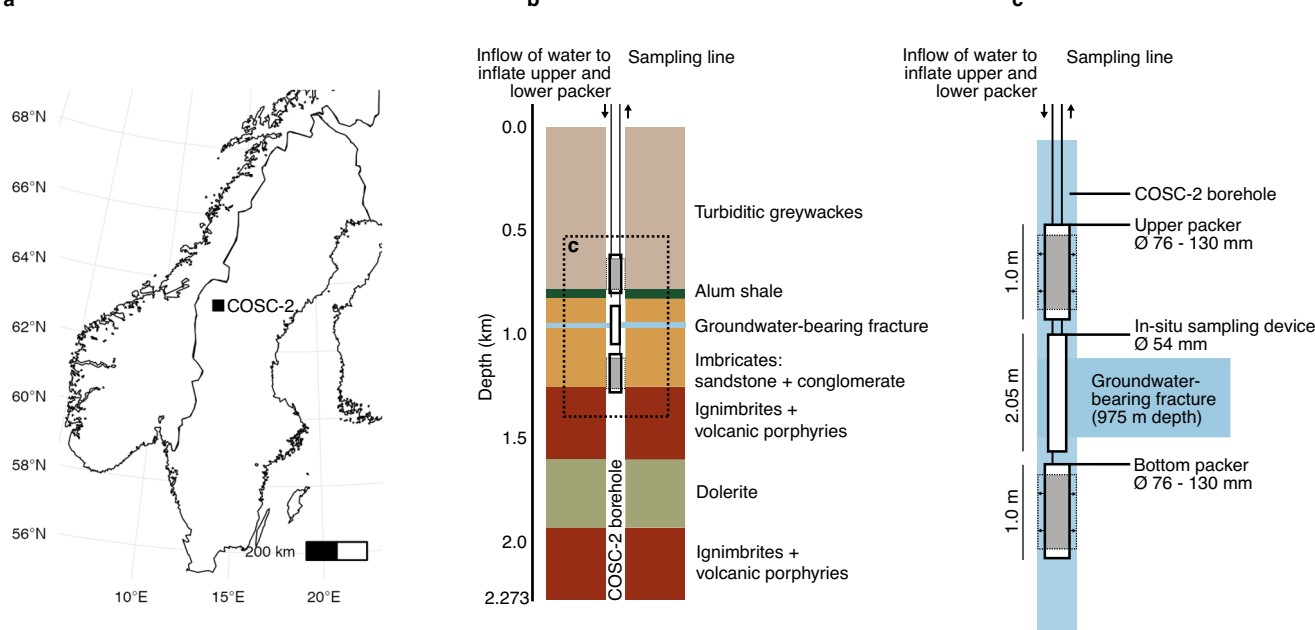

**Fig. 1 | Field site and groundwater collection using inflatable packers. a** Location of the COSC-2 borehole within Sweden (Lat 63.3124° N, Lon 13.5265° E). **b** Lithology of the borehole according to depth and the position of the packers used for isolating the groundwater-bearing fracture at 975 m below the surface (not drawn to scale). **c** Detail of the packers used during sampling. The packers (76 mm in diameter) were inflated against the borehole wall (96 mm in diameter) by pressurizing the rubber shell (gray surface) with water, thereby preventing to sample groundwater from other fractures intersected by the borehole. The samples for hydrochemistry were collected using both pumped water and an in situ sampling device that allowed water collection at the desired depth.

thermochronology models suggest that habitable temperatures (below 122 °C) prevailed during the last ~300 million years for the upper few kilometers of most parts of this craton[33]. Microbiological investigations on lower reaches of thick Paleozoic sedimentary piles that overlay the Precambrian strata in certain areas of the Fennoscandian shield, such in the Caledonian thrust sheets, are however more or less absent.

Highly isolated fluids at 755 m below sea level in the Paleozoic carbonate aquifer in the Death Valley (southeastern California), within Archaean metasediments at Mponeng, the Beatrix and Tau Tona gold mines in the Witwatersrand Basin (South Africa), as well as a borehole drilled into a Cretaceous aquifer in the Siberian artesian mega-basin have all been shown to hold a microbial community dominated by *Candidatus* Desulforudis audaxviator, with very low diversity[9,15,34]. This sulfate-reducing bacterium is of particular relevance for subsurface microbial ecology as it has a highly similar nucleotide identity (>99%) across continents (Africa, Eurasia, and North America) and has been hitherto exclusively described in deep subsurface groundwaters.

The COSC-2 (Collisional Orogeny in the Scandinavian Caledonides) borehole, located in central Sweden, was drilled through an upper 1250 m vertical section of early Paleozoic bedrock thrust sheets of the easternmost parts of the Caledonian orogen range that overlays deeper Precambrian crystalline bedrock (porphyries of ~1.8 Ga, and sections of dolerite intrusions) of the Fennoscandian Shield[35]. The drilling took place during spring to summer 2020 and extends 2250 m below the surface (elevation field site 320 m), thereby intersecting multiple natural, groundwater-bearing fractures. To accurately characterize this subsurface microbial diversity, it is desirable to isolate the groundwater-bearing fracture from other such fractures as these subsurface environments have been long-term undisturbed and to prevent the microbial communities to mix. It is also important to avoid intrusion of liquid from the main open borehole water pillar into the sampling connection, as this liquid is likely to contain contaminant microbial populations from the drilling operation[36]. One promising strategy to collect groundwater from a certain section in the borehole while reducing contamination from stagnant water in the main borehole, is to use inflatable packers to isolate the fracture of interest[37].

In this study, the microbial community in a groundwater-bearing fracture intersected by the COSC-2 borehole was characterized using metagenome sequencing combined with 16S and 18S ribosomal RNA gene amplicons. Microbial abundance was assessed using a combination of real-time PCR and flow cytometry. The main goal was to characterize this potentially long-term isolated subsurface microbial community in conjunction with characterization of the eukaryotic community.

## Results and discussion
### Groundwater characteristics

A groundwater-bearing fracture at 975 m depth located in a lower Paleozoic conglomerate rock section, intersected by the COSC-2 borehole, was isolated by employing inflatable packers (Fig. 1). The pH of the groundwater at this depth was 9.8 and the oxygen concentration in the range of 1.6 and 2.0 mg L$^{-1}$ ($n = 2$, Thomas Wiersberg, pers. comm.). Given the large depth of the borehole and the dominance of anaerobic bacterial populations, the low but detectable amount of oxygen indicated a possible contamination from either the drilling operation (drilled in 2020), sampling and sample treatment, or as a product of microbial activity[10]. The alkalinity was 24 mg CaCO$_3$ L$^{-1}$, the electrical conductivity 11.1 mS cm$^{-1}$, and the chloride to bromide mass ratio 142:1 (Cl$^-$ 3600 mg L$^{-1}$, Br$^-$ 57 mg L$^{-1}$). An overview of the hydrochemistry is provided in Supplemental Table 1. The groundwater intersected by the COSC-2 borehole most resembled groundwaters from Forsmark (Sweden), recharged by brackish glacial meltwater, at a depth of 409 to 549 m below sea level[38] (Supplemental Fig. 1). In contrast, the pH, chloride and magnesium concentrations, and the chloride to bromide mass ratio from the COSC-2 groundwater differed from groundwaters intersected by boreholes at the Paleoproterozoic granitoid-hosted Äspö Hard Rock Laboratory (Sweden) at depths between 69 and 467 m[30,31], and groundwaters intersected by Outokumpu borehole (Finland) at 500, 1000, and 1500 m depth (hosted by wall rocks Proterozoic mica schist and black schist)[29]. Hence, comparison of this and other deep subsurface groundwaters located in the Fennoscandian Shield, based on chloride and magnesium concentration and the chloride to bromide mass ratio, indicated a non-marine origin of the studied groundwater.

**Table 1 | Details of groundwater sampling**

| Filter | Groundwater filtered (L) | DNA yield (ng L$^{-1}$) | Bacterial gene copies (mL$^{-1}$) | Archaeal gene copies (mL$^{-1}$) | No. ASVs / bins |
|---|---|---|---|---|---|
| 1 + 2 | 4.1 | 20.1 | $3.3 \times 10^4 \pm 153$ | $27 \pm 1.2$ | 177/24 |
| 3 + 4 | 2.9 | 11.9 | $1.7 \times 10^4 \pm 1.1 \times 10^3$ | $27 \pm 3.6$ | 134/26 |
| 5 + 6 | 2.6 | 8.2 | $1.5 \times 10^4 \pm 523$ | $24 \pm 1.6$ | 218/25 |
| 7 | 0 (control) | < 0.05 | BD | BD | 43 |

Groundwater samples were collected and filtered on 3 September 2022. Two filters were combined during DNA extraction to increase the yield, except for the negative control sample. The DNA yield is presented as the amount of DNA extracted standardized by the volume of groundwater filtered (ng L$^{-1}$). The reported bacterial and archaeal abundances (mean ± sd, $n = 3$) were quantified using qPCR, targeting the 16S rRNA gene. The number of gene copies in the control sample were below detection limit (BD; ΔCq with no-template controls below 3).

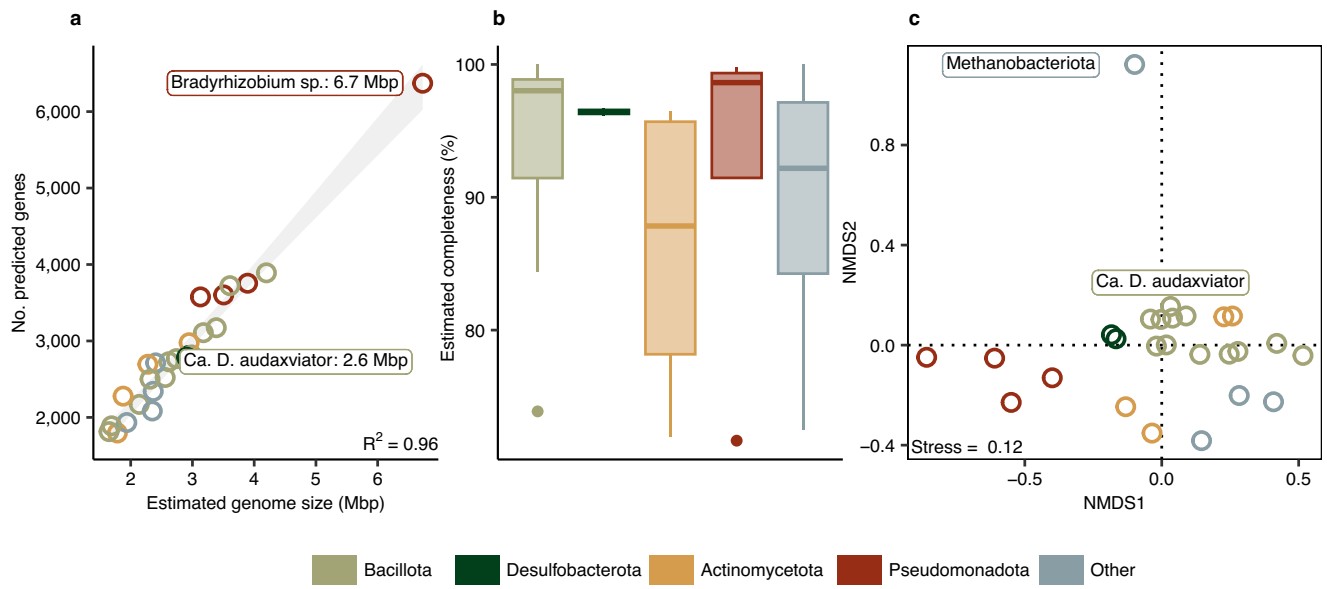

**Fig. 2 | General characteristics of the de-replicated genomes. a** Number of predicted genes versus estimated genome size showing a high association ($R^2 = 0.96$). **b** Estimated genome completeness as quantified using 104 marker genes. Out of a total of 26 de-replicated genomes, eighteen had a completeness above 90% and eight had a completeness between 70 and 90%. **c** Divergence among genomes based on functional orthologous genes (KO functional orthologs, accounting for multiple gene copies) using non-metric dimensional scaling (NMDS).

## Sequencing output

To characterize the subsurface community, cells from a total of 9.6 L groundwater were captured by membrane filtration ($n = 6$, pore size 0.1 µm). Two membranes were pooled during DNA extraction, resulting in three DNA extracts that contained 20, 12, and 8 ng L$^{-1}$ genomic DNA per volume groundwater filtered (Table 1). Sequencing of 16S rRNA gene amplicons produced a total of 257 amplicon sequencing variants (ASVs) from $78 \times 10^3$ sequencing reads (standard deviation $23 \times 10^3$, $n = 3$, Supplemental Table 2). In addition, sequencing of the 18S rRNA gene produced a total of 2982 ASVs from $136 \times 10^3$ (sd $43 \times 10^3$, $n = 3$). Rarefaction curves were asymptotic and clearly demonstrated sufficient sequencing depth to capture the diversity in the sequencing libraries (Supplemental Fig. 2). Contamination during either sample collection, DNA extraction, or DNA amplification is a major concern as biomass in the deep subsurface is usually several orders of magnitude lower compared to samples retrieved from surface environments[39]. The DNA concentration of the negative control collected at the field site (blank filter) was below detection limit (0.05 ng µL$^{-1}$) and PCR amplification did not yield a visible product during gel electrophoresis. Nevertheless, this library was sequenced and featured 42 ASVs from a total of 2183 reads. The community composition in the control sample did not resemble the community in the samples of interest (Supplemental Fig. 2). Only two ASVs were overlapping between the communities and those were removed from downstream analyses.

The metagenomes ($n = 3$) contained a minimum of 16.1 and a maximum 25.2 million paired-end reads and produced 26 unique de-replicated

metagenome-assembled genomes (MAGs; hereafter termed reconstructed genomes). All were bacterial except one archaeal (Fig. 2). Eighteen genomes had an estimated completeness above 90% and the remaining eight had an estimated completeness between 70 and 90%[40].

## Microbial abundance

Bacterial abundance (Table 1) was $22 \times 10^3$ 16S rRNA gene copies mL$^{-1}$ (sd $9.0 \times 10^3$, $n = 18$) while the archaeal abundance was much lower with only 26 gene copies mL$^{-1}$ (sd 2.4, $n = 18$). Based on flow cytometry, the microbial abundance was estimated to be in the range of 22.4 and $22.8 \times 10^3$ cells mL$^{-1}$ ($n = 2$). These numbers were in the same range ($1.3 \times 10^4$ to $4.4 \times 10^5$ cells mL$^{-1}$) as previous epifluorescence microscopy-based estimates from a groundwater-bearing fracture at 967 m depth, intersected by the Outokumpu borehole, also located in the Fennoscandian Shield[37]. In general, multiple studies on the continental deep subsurface report cell density to be in the order of $10^4$ cells mL$^{-1}$ groundwater[12,30,41].

## Microbial community structure

The subsurface community was dominated by the Bacillota (synonym Firmicutes), with 12 of the 26 reconstructed genomes being affiliated with this phylum and accounting for 88% of the metagenomic read coverage (relative abundance was expressed as percentage read coverage; Fig. 3). More specifically, *Candidatus* Desulforudis audaxviator and *Desulfitibacter* sp. (both representatives of the Bacillota) were highly abundant and

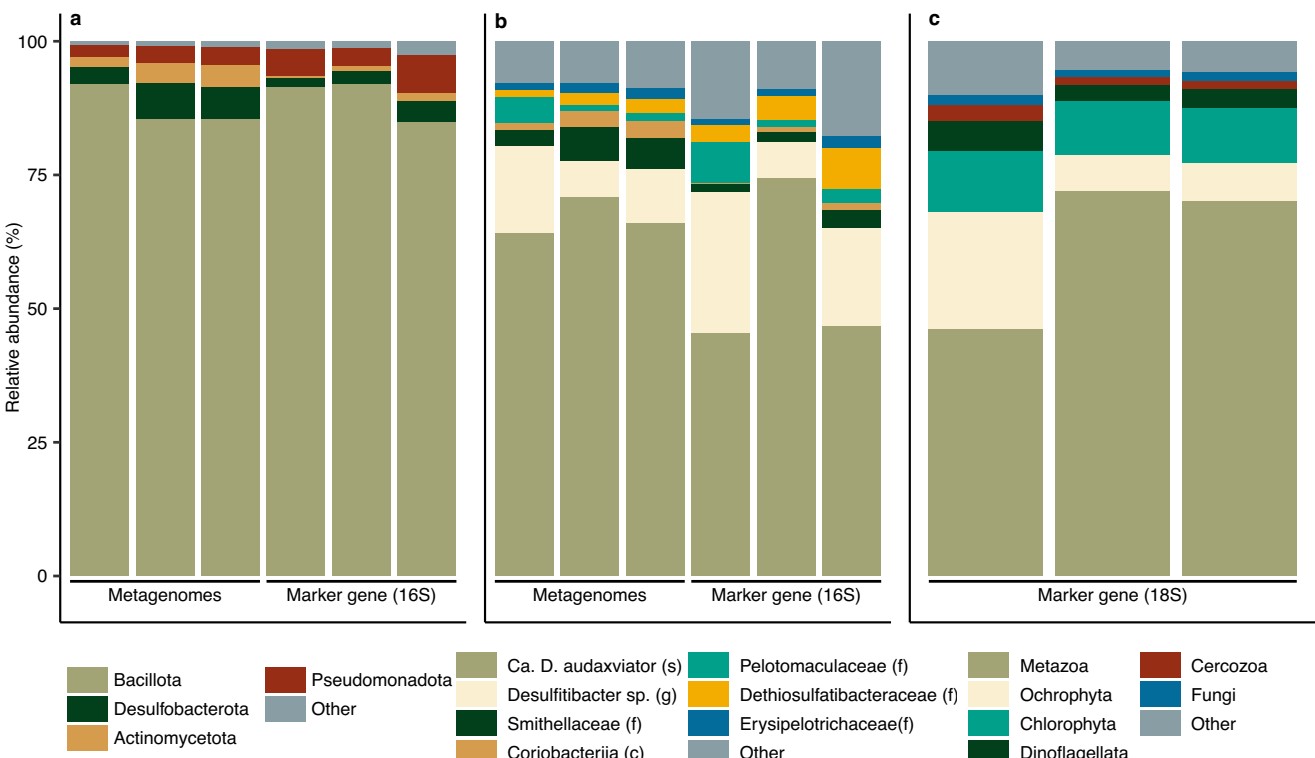

**Fig. 3 | Overview of the microbial community structure. a** Bacterial community composition depicted at the level of phylum. Bacillota (synonym Firmicutes) dominated both the metagenomes and the marker-gene dataset in terms of relative abundance. Phyla are arranged according to abundance while grouping low-abundant phyla as "Other". **b** Bacterial community composition depicted at the highest taxonomic resolution possible, i.e., class (c), family (f), genus (g), or species (s). **c** Eukaryotic community composition based on 18S rRNA gene amplicons, depicted at the level of phylum. Phyla are arranged according to abundance while grouping low-abundant phyla as "Other".

accounted for 78% of the metagenomic read coverage. The remaining reconstructed genomes were affiliated with the phyla Desulfobacterota (5.4% read coverage), Actinomycetota (synonym Actinobacteria; 3.4%), Pseudomonadota (synonym Proteobacteria; 2.9%), Chloroflexota (0.4%), Bacteroidota (0.4%), and Methanobacteriota (0.1%). The community structure according to the metagenomes strongly resembled the community based on the PCR-based 16S ribosomal RNA although no archaea were detected in the latter assay. This implied that the primers used adequately captured the bacterial subsurface diversity in this groundwater even if the primers were originally designed for marine ecosystems. It is not unusual for populations affiliated with the Bacillota to be abundant or even dominate groundwater communities in the deep subsurface[3,9,15] and this has been associated with the long residence time of the groundwater with the ability of Bacillota to thrive in bedrock fractures isolated from recent surface recharge[3]. Together with the stability (low tectonic activity), hydrochemistry, and age (Precambrian) of the studied bedrock[28], the dominance of Bacillota suggested this groundwater-bearing fracture at 975 m depth to be long-term isolated from surface recharge.

The taxonomic resolution of the eukaryotic ASVs (according to 18S rRNA gene amplicon sequencing) was similar to the prokaryotic ASVs, with 91% of the eukaryotic ASVs characterized on the level of phylum, 82% on order, and 53% on genus compared to 94, 83, and 59% for the prokaryotic ASVs, respectively. Furthermore, the eukaryotic community comprised 297 eukaryotic genera while 126 prokaryotic genera were detected. Whether this eukaryotic diversity originated from surface water recharge or leached from the soil was unclear. The eukaryotic community was dominated by the flatworm *Praeconvoluta castinea* that was responsible for 45% of the amplicon read counts (Fig. 3). Other abundant groups were the chlorophyte *Desmodesmus* sp. (6.9%), the crustacean *Acartia tonsa* (6.6%), and the ochrophyte *Spumella elongata* (3.7%). Borgonie et al.[18]. demonstrated the presence of eukaryotic groups such as flatworms, fungi, and arthropods in the

fracture waters on the deep subsurface and suggested that food instead of dissolved oxygen was the limiting factor for growth. Additional studies found fungi and chlorophytes to be abundant in deep subsurface groundwaters[42]. Thus, all the major eukaryotic groups presented here have been detected before in groundwaters of the deep subsurface. A considerable proportion of the eukaryotic community presumably respire aerobically (for example, *Desmodesmus* sp. or *Acartia tonsa*) and their origin and function in the groundwater under scrutiny here is largely unknown. Possibly, some eukaryotic groups originated from the main borehole pillar, despite using packers for a more targeted groundwater retrieval. However, as the eukaryotic community was reconstructed from the same DNA sample as the prokaryotic community, one would expect aerobic clades also to be abundant in the latter community while the abundance of aerobic prokaryotes was rather low.

Fungi were also detected and were mainly represented by the Cryptomycota (1.2%), Oomycota (0.8%), and Chytridiomycota (0.4%). ASVs affiliated with these fungal clades ($n = 193$) were, in addition to the PR² database[43], aligned with entries in the RefSeq NCBI database[44] but did not result in additional taxonomic delineation. Previously the Cryptomycota have been detected in the continental deep biosphere where they can inhabit sedimentary rock matrices at around 250 m depth[42] and subsurface groundwaters at around 10 m depth[45]. Literature on this fungal clade in the deep subsurface in general and deep fracture fluids is either scarce or absent, respectively. This phylum was only recently discovered using a genetic approach and the Cryptomycota are unique as fungal representatives in lacking the chitin-rich cell wall[46]. Representatives of the Cryptomycota are described in the context of an endo- or epibiotic lifestyle with the potential host affiliated with the phyla Oomycota or Chytridiomycota[47]. The presence of the Cryptomycota, Oomycota, and Chytridiomycota in this groundwater-bearing fracture at 975 m depth hints at a putative unrecognized role of fungi as potential decomposers of refractory organic matter in the continental deep biosphere[26,27].

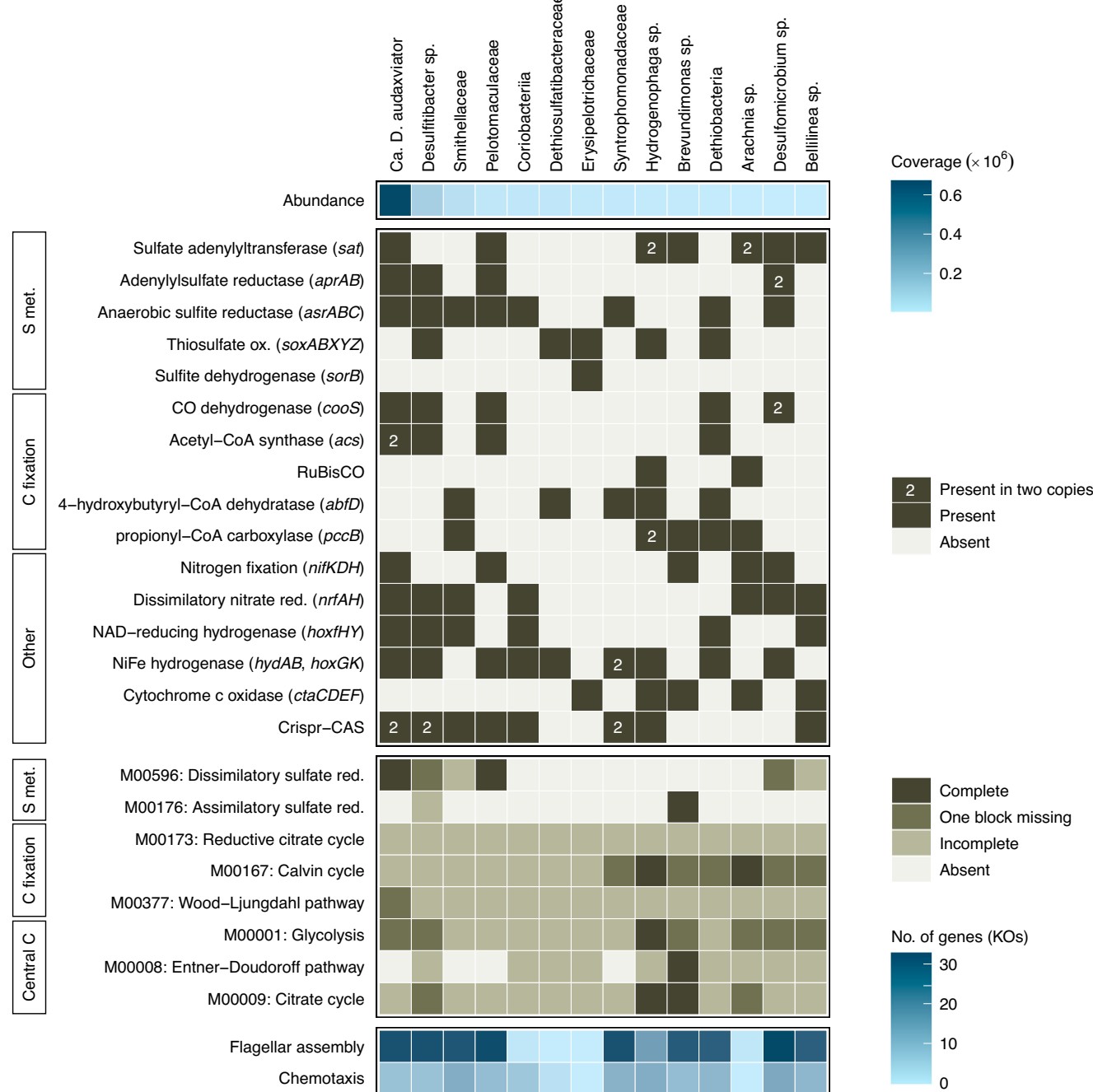

**Fig. 4 | Genomic potential of the most abundant clades.** Included in the heat map are (from top to bottom) abundance, genomic potential based on marker genes (present / absent), completeness of selected pathways (the module identifier referring to KEGG modules), and pathways of flagellar assembly and chemotaxis (no. of gene orthologs). Only genomes with a metagenomic read coverage above 1% are included. The EC references for the marker genes are provided in the method section.

## Genomic potential

A substantial portion of the reconstructed genomes contained genes involved in carbon fixation (9 out of 26 genomes, metagenomic read coverage 85%), nitrogen fixation (6, 72%), hydrogen oxidation (16, 94%), sulfate reduction (3, 72%), and sulfur oxidation (14, 93%).

Both the presence of individual marker genes and pathway completeness in the reconstructed genomes were used to identify how the detected subsurface populations could fix carbon. Of the major fixation pathways, the reductive acetyl-CoA (Wood-Ljungdahl) pathway was the most prevalent (Fig. 4), and the genomes of the two most abundant populations (together accounting for 78% metagenomic read coverage) contained carbon monoxide dehydrogenase (*cooS*) and acetyl-CoA synthase (*acsA*). The Wood-Ljungdahl pathway is reported to be used by prokaryotes living in strict anoxic, energy-limited environments that are close to the thermodynamic limit of life, and is observed in e.g., sulfate-reducing bacteria, acetogenic bacteria, and methanogenic archaea[24]. Propionyl-CoA carboxylase (key enzyme of the 3-hydroxypropionate bi-cycle) was also abundant. This cycle differs from the Wood-Ljungdahl pathway in that it uses bicarbonate as a carbon source instead of carbon dioxide and also has a higher energic cost while requiring less metals and coenzymes[24,48]. Additionally, 4-hydroxybutyryl-CoA dehydratase (key enzyme of the dicarboxylate-hydroxybutyrate cycle) was detected in multiple genomes. This enzyme is a key enzyme in the oxygen-insensitive hydroxypropionate-hydroxybutyrate cycle; however, comparing pathway

completeness revealed that it was more likely that the oxygen-sensitive dicarboxylate-hydroxybutyrate cycle was used. The latter cycle, together with the 3-hydroxypropionate bi-cycle, are described as using bicarbonate as an inorganic carbon source and this may be beneficial under the prevailing alkaline conditions[24]. Regarding the reductive citrate cycle, acetyl-CoA is formed from $CO_2$ and the further conversion of acetyl-CoA requires three enzymes (fumarate reductase, 2-oxoglutarate oxidoreductase, and ATP-citrate lyase) that are characteristic of this cycle[49]. As only two out of three of these enzymes were detected, with ATP-citrate lyase not present in any of the genomes, the fumarate reductase and 2-oxoglutarate oxidoreductase were most likely involved in the conventional citrate cycle rather than the reverse cycle[49].

The observed completeness of the carbon fixation pathways was low. Despite high estimated genome completeness with on average 96% (sd 7.0, $n = 14$) for the more abundant populations included in this analysis (read coverage > 1%), it was challenging to robustly determine whether the low pathway completeness was due to genome completeness or rather due to a reduced genome size[50]. Of all the major carbon fixation pathways included in this analysis, only two genomes (*Arachnia* sp. and *Hydrogenophaga* sp.) contained all the genes comprising one of these pathways, i.e., the Calvin cycle. Interestingly, aside from containing all the genes for the energy-demanding Calvin cycle, the genome of *Hydrogenophaga* sp. also contained the aerobic type of carbon monoxide dehydrogenase (*cutSML*) and the key enzymes of the 3-hydroxypropionate bi-cycle (*abfD*) and the dicarboxylate-hydroxybutyrate cycle (*pccAB*). This genus is repeatedly detected in subsurface groundwaters[3,10,29] and cultured representatives have been described to be both chemoorganotrophic and chemolithotrophic[51]. The genome of *Hydrogenophaga* sp. was different from the majority of reconstructed genomes as it contained several genes (RuBisCO, *cutSML*, and *abfD*) involved in multiple carbon fixation pathways. This was unlike most genomes, which encoded only one carbon fixation pathway (Fig. 4). Having the genomic potential for aerobic respiration (NADH ubiquinone oxidoreductase, the cytochrome $bc_1$ complex, and cytochrome $c$ oxidase) and aerobic carbon fixation suggested that *Hydrogenophaga* sp. could act as an oxygen scavenger coupled to hydrogen oxidation, while having the genomic potential for thiosulfate oxidation to sulfate (*glpE*, *sox*), and oxidation of sulfur to sulfite (*sdo*) in the absence of oxygen. In general, based on the presence of key enzymes, and especially when taking the abundance of the individual populations into account, the Wood-Ljungdahl pathway was the most prevalent carbon fixation pathway in the community, supporting the trend that this pathway is selected for in low-energy ecosystems such as the continental deep subsurface.

Regarding nitrogen cycling, genes involved in nitrogen fixation (*nifDHK*) and dissimilatory nitrate reduction to ammonium (*nrfADH*) were prominent in abundant populations affiliated with the Bacillota. Genes coding for nitrification (*amoABC*) and anammox (*hzoA*, *hzsA*) were not detected while those encoding for nitrate and nitrite reduction (*napAB*, *narGH*, *nirKS*, *nirBD*) were solely detected in low-abundant genomes affiliated with the Pseudomonadota and Actinomycetota. This suggested that populations affiliated with the Bacillota mediated nitrogen fixation and dissimilatory nitrate reduction to ammonium (DNRA) in this subsurface groundwater, while denitrification was mainly encoded by less abundant populations affiliated with the Pseudomonadota and the Actinomycetota.

For sulfur cycling, 14 out of 26 reconstructed genomes (read coverage 93%) encoded genes involved in sulfur metabolism. These populations were affiliated with Bacillota (eight reconstructed genomes), Pseudomonadota (4), and Desulfobacterota (2). The high abundance of populations potentially involved in sulfate reduction (*aprAB*, *sat*) and the presence of populations containing genes involved in sulfur oxidation (*sdo*, *sor*) suggested that sulfur cycling could occur in this groundwater. Sulfur isotope measurements on the dissolved sulfate (0.5 mM, total sulfur pool 1.0 mM) in the groundwater yielded $\delta^{34}S$ value of 12.4‰ VCDT (sd 0.14) that was similar or slightly lower (by 10–20‰) compared to Paleozoic marine sulfate $\delta^{34}S$[52], meaning that the sulfate may originate from such connate waters.

### *Candidatus* Desulforudis audaxviator

Based on the metabolic weight score, a metric that takes the genomes' metabolic potential and coverage into account, *Ca.* Desulforudis audaxviator contributed to most of the metabolic functions related to sulfur cycling, carbon fixation, and nitrogen fixation (Supplemental Fig. 3). To date, *Ca.* Desulforudis audaxviator has been exclusively detected in the continental deep subsurface and has been reported to dominate fracture waters at ~3 km depth in the Mponeng (pH 8.6) and Tau Tona (pH 8.6) gold mines (South Africa)[3,15], a 2-km-deep groundwater (pH 8.2) in Siberia[34], and a groundwater (pH 8.2) in North America at 750 m depth[9]. The success of the sulfate-reducing lineage *Ca.* Desulforudis audaxviator in these deep subsurface and slightly alkaline ecosystems has been ascribed to the ability to switch between an autotrophic and heterotrophic lifestyle with concomitant potential to fix nitrogen[15]. Indeed, its genome (estimated to be 2.6 Mbp in size, GC content 60.9%, 98% complete, 2728 predicted genes) contained the key genes necessary for dissimilatory sulfate reduction, nitrogen fixation, autotrophic carbon fixation, and flagellar assembly (Fig. 4). The genome also contained two clustered regularly interspaced short palindromic repeat (CRISPR) and adjacent CRISPR-associated genes, potentially involved in viral defense. Comparison of the average nucleotide identity (ANI) of the reconstructed genomes generated in this study ($n = 2$) with those from South Africa[3,6,53] ($n = 7$) and Russia[34] ($n = 1$) resulted in an average ANI of 98.8% (sd 0.22, minimum 98.3, $n = 10$; Fig. 5) as compared to 99.5% (sd 0.35, minimum 98.8, $n = 8$) when omitting the reconstructed genomes from the present study. These results suggested that the reconstructed genomes generated with this study are less similar compared to those from South Africa and Russia. However, it should be noted that the variations in DNA extraction methods possibly influenced this relatively low average ANI compared to the minimum of 99.2% reported by Becraft et al.[9].

## Conclusion

The genome-resolved metagenomics revealed a predominantly bacterial community with a microbial abundance typical for low-energy, deep subsurface groundwaters and mainly consisting of Bacillota (88% of the metagenomic read counts). The Bacillota were largely represented by the sulfate-reducer *Ca.* Desulforudis audaxviator and contained genes enabling survival under anaerobic, low-energy conditions, such as the carbon monoxide dehydrogenase as part of the Wood-Ljungdahl pathway for the fixation of inorganic carbon. *Ca.* Desulforudis audaxviator being abundant in deep subsurface, anoxic groundwaters has been reported in several independent studies, implicating this taxon as well-adapted to low-energy conditions. Finally, the high diversity of the eukaryotic community, including presumably aerobic populations, suggested an unexplored role of prokaryotes in this subsurface ecosystem.

## Methods
### Groundwater sampling

The COSC-2 borehole (Lat 63.3124° N, Lon 13.5265° E) has a depth of 2276 m from the surface (elevation 320 m) and was drilled in clastic sedimentary rocks and the crystalline basement of the Fennoscandian Shield. A groundwater-bearing fracture within a lower Paleozoic conglomerate rock section at 975 m depth was isolated from other fractures using inflatable rubber and stainless-steel packers (Lapela Technology Oy, Rauma, Finland) positioned below and above the fracture (Fig. 1). The temperature in the borehole at this depth was 25 °C (data from Lorenz et al.[54]). The packers (76 mm in diameter) were inflated against the borehole wall (96 mm in diameter) using a 4 mm polyamide tube (Toppi Oy, Espoo, Finland) that was pressurized using a manual pressure pump filled with tap water. These packers were operated with a custom-made winch trailer (Lapela Technology Oy) and secured with a fiber rope 5 mm in diameter. Samples for the hydrochemistry were taken both from pumped fluid (pH, alkalinity, salinity, sulfur isotope) and from a custom-made in situ fluid sampling device (Lapela Technology Oy) placed below the upper packer. Electrical conductivity and pH were measured in the field using a portable device (WTW). Alkalinity was determined the same day by end-point titration to pH 4.5

**Fig. 5 | Genome comparison of *Ca*. Desulforudis audaxviator.** The reconstructed genomes (*n* = 10) were compared based on average nucleotide identity. Rows were clustered according to Euclidean distances. Included genomes originated from Chivian et al. (2008)[15], Lau et al. (2014)[53], Magnabosco et al. (2016)[3], Lau et al. (2016)[6], and Kadnikov et al. (2018)[66]. SA; South Africa.

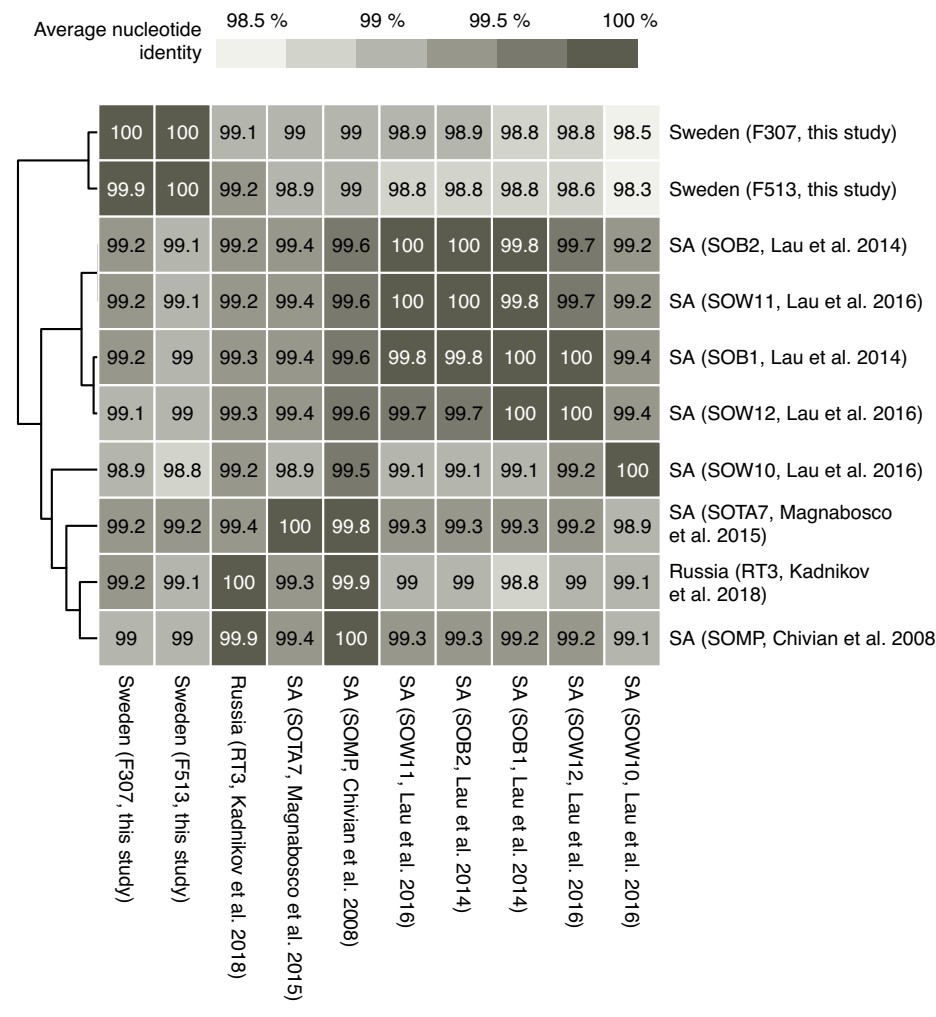

with 0.16 M sulfuric acid using a digital titrator (Hach, model 16900). Gas samples were taken from the in situ sampling device and gas analyses were conducted at GFZ Potsdam using a Quadrupole Mass Spectrometer (QMS, OmniStarTM GSD 300, Pfeiffer Vacuum, Germany). Elemental analyses were done with inductively coupled plasma mass spectrometry (Neptune MC-ICP-MS; Thermofisher). Anions (Br, Cl, F, $SO_4$, and $NO_3$) were analyzed with ion chromatography. All geochemical laboratory analyses were conducted by an external laboratory (Eurofins Environment Testing Finland Oy, Lahti, Finland). Sulfur isotopes ($^{32}S$ and $^{34}S$) were analyzed on a high-resolution multicollector (MC-ICP-MS; Nu Instruments). Ion exchange was performed with a cation exchange resin (AG 50W-X8, 200 to 400 μm mesh). Sulfur isotope analyses were conducted by the Geological Survey of Finland. Prior to sampling, the sampling connections, including ca. 1000 m of polyamide tubing 10 mm in diameter, were flushed overnight with three section volumes (3 × 314 L) to avoid contamination from stagnant water. Pumped water samples for anions and alkalinity (2 × 250 mL) were unfiltered and subsampled (60 mL) for the sulfur isotope analysis. This subsample was filtered with a 0.45 μm pore size. Pumped water samples for cations (100 mL) were filtered (0.45 μm pore size) and acidified with 0.5 mL of 65% ultrapure $HNO_3$. Cells were captured under a flow rate of 1.5–2.1 L h$^{-1}$ using a high-pressure filter holder (Millipore) containing a sterile polyvinylidene fluoride membrane with a 0.1 μm pore size (Merck Durapore, diameter 47 mm). The membranes (*n* = 7) were aseptically collected in extraction tubes and immediately frozen in a mobile −80 °C freezer that was also used to transport the samples to the laboratory where the samples were stored at −80 °C. The side of the filter containing the cells (facing the water source) faced inwards in the collection tube to maximize yield during DNA extraction. Samples for flow cytometry (10 mL, *n* = 2) were fixed in 2.2% (vol/vol) formaldehyde on site and stored at 4 °C.

## DNA extraction and amplification

Extraction was performed in a room dedicated to the extraction of nucleic acids while working in a fume hood that was cleaned with both bleach and 70% ethanol prior to use. All plastic consumables (tips, centrifuge tubes, multiwell plates) were PCR grade and UV treated for 10 min before placing the material in the fume hood. DNA was extracted using the DNeasy PowerWater kit (Qiagen), following the manufacturer's protocol apart from eluting the nucleic acids in 50 μL instead of 100 μL of the included elution buffer. For every extraction, two filters were pooled to increase the DNA yield, except for the control sample. The V3-V4 region of the 16S rRNA gene was amplified using the primer pair 341 F and 805R[55] while the 18S rRNA gene (V4-V5 region) was amplified with the 454 F 981 R primer pair[56]. The product of the first PCR served as template for a second amplification step containing the sample-specific sequencing barcodes. In total, the PCR programs contained 32 and 37 cycles for the 16S and 18S assays, respectively. The amplified product was purified after each PCR using the AMPure XP reagent (Beckman Coulter). To identify potential contaminants introduced during field sampling or molecular work, a blank filter was collected at the sampling site and processed simultaneously with the other samples. Prior to equimolar pooling, the concentration and fragment length distribution of the amplified products were measured using a Qubit 2.0 fluorometer (Life Technologies) and automated gel electrophoresis (Agilent TapeStation 4150), respectively. Sequencing was done

at the Science for Life Laboratory, Sweden on an Illumina MiSeq (v3-600), producing 2 × 301 bp paired-end reads.

Libraries for metagenomic sequencing were prepared using the Tecan MagicPrep while adjusting the cycling conditions (10 to 12 cycles) to the amount of template DNA added (10 to 20 ng) according to the manufacturer's instructions. The concentration of the libraries was measured using a Qubit 3 fluorometer (Life Technologies). After equimolar pooling, the fragment length distribution of the pooled libraries was checked using agarose gel electrophoresis. The metagenomes were sequenced at Science for Life Laboratory (SNP&SEQ platform) on an Illumina NovaSeq platform equipped with a SP flowcell, producing 2 × 150 bp paired-end reads.

### Real-time PCR and flow cytometry

Microbial abundance was assessed according to 16S rRNA gene copies using a real-time PCR (qPCR) on a LightCycler 480 (Roche Diagnostics). The reaction volume was 10 μL and consisted of 5 μL Platinum SYBR Green qPCR SuperMix-UDG with ROX (Thermo Fisher Scientific), 0.4 μL 10 μM primer (Eurofins), 3.2 μL nuclease-free water (Thermo Scientific), and 1.0 μL template. Bacterial and archaeal gene fragments were amplified using the primers 908F_mod/1075R[57] and 915 F/1059R[58], respectively. Cycling conditions were 2 min at 95 °C, 40 cycles of 15 s at 95 °C and 30 s at 60 °C, followed by a melt curve analysis to assess primer specificity. Standard curves were generated with a dilution series of purified PCR product using genomic DNA of pure cultures as template (i.e., *Acidiphilium cryptum* JF-5 for bacteria and *Ferroplasma acidiphilum* BRGM4 for archaea). Standards ($n = 7$), samples of interest ($n = 3$), and no-template controls ($n = 1$) were run in triplicates and the former two were 1:10 and 1:100 diluted in nuclease-free water to account for inhibition of the polymerase. The gene copy numbers were reported in gene fragments mL$^{-1}$ after correcting for groundwater volume filtered for DNA extraction and the elution volume. Reaction efficiency for the bacterial and archaeal assays were 89.9% and 87.5%, respectively. A minimum number of three quantification cycles ($\Delta$Cq) between samples and no-template controls was maintained as a limit of detection.

In addition to qPCR, microbial abundance was also assessed using a flow cytometer (Cytoflex, Beckman Coulter) equipped with two lasers (blue and violet) and a microwell plate autosampler. The fixed samples (2.2% formaldehyde, $n = 2$) were stained using SYBR Green I that binds to DNA, thereby visualizing both heterotrophic and autotrophic populations. Sterile-filtered ultrapure water (Milli-Q, pore size 0.2 μm) was used as a reference to distinguish cells of interest from any background signal.

### Bioinformatics

Raw sequencing reads from the 16S rRNA gene amplicon ($n = 4$) were processed using the ampliseq pipeline[59] (v2.3.0) from the nf-core framework that relied on Nextflow (v21.10.6), Cutadapt (v3.4), FastQC (v0.11.9), DADA2 (v1.22.0), and the SBDI Sativa curated 16S GTDB database[60] (release 207). The ampliseq pipeline was run with default settings, except for the trimming of the primers whereby reads not containing the primer sequence or containing double copies were discarded from downstream analysis. The raw reads from the 18S rRNA gene amplicon ($n = 3$) were processed identically to the 16S amplicons, except from using the Protist Ribosomal Reference database[43] (v4.14.0) for taxonomy assignment.

Raw sequences from the metagenomes ($n = 3$) were assembled, binned, and annotated on default settings if not specified using the curated mag pipeline[61] (v2.2.1) from the nf-core framework[62]. Adapter plus quality trimming was done with fastp (v0.23.2), reads were co-assembled by grouping the three metagenomes using MEGAHIT (v1.2.9), followed by evaluation of the contigs with Quast (v5.0.2). Protein-coding genes were predicted using Prodigal (v2.6.3), open reading frames were functionally annotated with Prokka (v1.14.6), and the metagenomes were binned with MetaBAT2 (v2.15). The bins were refined with DAS Tool (v1.1.4) after which the quality of the binned genomes was evaluated with CheckM (v1.1.3). dRep (v3.4.2) was used for de-replication, setting the maximum contamination to 5% and the minimum completeness to 70%, followed by

taxonomic assignment with GTDB-Tk (v2.1.1). The read coverage of the bins was normalized to transcripts per million (tpm) after mapping the quality-trimmed reads from the three metagenomes (the ouput of fastp) to the de-replicated bins with CoverM (v0.6.1). The de-replicated genomes were additionally functionally annotated with eggNOG-mapper (v2.1.9) and the obtained KEGG orthologs were used to identify the completeness of KEGG modules (KEGG mapper) by scoring them as complete, one block (gene) missing, incomplete, or absent. The presence of genes involved in viral defense was checked using the CRISPRCasFinder[63], setting the minimum evidence level to 3 and requiring both CRISPR and CRISPR-associated genes to be present.

Marker genes included for constructing the heat map on genomic potential were sulfate adenylyltransferase (*sat*; EC:2.7.7.4), adenylylsulfate reductase (*aprAB*; EC:1.8.99.2), anaerobic sulfite reductase (*asrABC*; K16950, K16951, K00385), L-cysteine S-thiosulfotransferase (*soxABXYZ*; EC:2.8.5.2), sulfite dehydrogenase (*sorAB*; EC:1.8.2.1), nitrogenase (*nifDHK*; EC:1.18.6.1), nitrite reductase (*nrfAH*; EC:1.7.2.2), anaerobic carbon monoxide dehydrogenase (*cooS*; EC:1.2.7.4), CO methylating acetyl-CoA synthase (EC:2.3.1.169), ribulose-bisphosphate carboxylase (rubisco; EC:4.1.1.39), 4-hydroxybutanoyl-CoA dehydratase (*acsA*; EC:4.2.1.120), propionyl-CoA carboxylase (*pccAB*; EC:6.4.1.3), hydrogen dehydrogenase (*hoxHY*; EC:1.12.1.2), NiFe hydrogenase (*hoxGK*; EC:1.12.99.6, *hydAB*; EC:1.12.2.1), and cytochrome *c* oxidase (*ctaCDEF*, EC:1.9.3.1). The functional annotation from both Prokka and eggNOG-mapper were queried based on the EC reference or gene name and the output was manually curated to verify if the gene of interest was obtained. The de-replicated reconstructed genomes combined with the quality-trimmed reads (the output of fastp) were analyzed using METABOLIC[64] (v4), thereby producing the metabolic weight scores for each reconstructed genome. Finally, the genomes affiliated with *Ca.* Desulforudis audaxviator were compared with reconstructed genomes reported in literature based on average nucleotide identity using FastANI[65] (v1.34).

### Statistics and reproducibility

Statistics and data visualization were performed in R (v4.2.1). For the amplicon data, absolute counts were standardized within a sample to relative abundances by dividing an ASVs count by the total number of counts within a sample.

### Reporting summary

Further information on research design is available in the Nature Portfolio Reporting Summary linked to this article.

### Data availability

All sequencing data has been made publicly available at the European Nucleotide Archive under project reference PRJEB68186. A compiled version of the R Markdown document, together with the number of sequence reads (16S and 18S rRNA gene) throughout the bioinformatic pipeline, are provided on GitHub at https://doi.org/10.5281/zenodo.13882042.

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

## Acknowledgements

The authors acknowledge Henning Lorenz from Uppsala University and Arto Pullinen from the Geological Survey of Finland for their support during the field sampling, Emelie Nilsson for her advice and support for the bioinformatic processing, and Maliheh Mehrshad for running the flow cytometry. Tabo V. Jain is acknowledged for his help in preparing the 18S sequencing library. Matthias Hoetzinger and Prune Leroy are acknowledged for their help in preparing the metagenomic sequencing libraries. The study was supported by The Swedish Research Council (Vetenskapsrådet contracts 2018-04311 to MD plus 2017-05186 and 2021-04365 to HD), FORMAS (2020-01577 to HD), Olle Engkvist's Foundation (contract 216-0462 to MD), the Crafoord foundation (contract 20210524 to HD), J. Richt. Foundation (to HD). SB acknowledges financial support from the Swedish Research Council and Science for Life Laboratory. High-throughput sequencing was carried out at the National Genomics Infrastructure hosted by the Science for Life Laboratory. Bioinformatics analyses were carried out utilizing the Uppsala Multidisciplinary Center for Advanced Computational Science (UPPMAX) at Uppsala University (projects NAISS 2023/22-893 & 2023/6-261). The computations were enabled by resources provided by the Swedish National Infrastructure for Computing (SNIC) at UPPMAX partially funded by the Swedish Research Council through grant agreement no. 2016-07213. Funding for open-access publishing provided by Linnaeus University.

## Author contributions

H.D., M.D., S.B., and G.W. designed the research; H.D., F.D., G.W., and R.K. performed field work; R.K. performed hydrochemistry measurements; G.W. and F.D. performed the molecular work; G.W. and C.G.R processed and performed ecological interpretations of the sequencing data; G.W. and M.D. wrote the manuscript with comments from all authors. M.D. and H.D. provided funding. All authors read and improved the final version.

## Funding

## Competing interests

The authors declare no competing interests.
