## [Peer Review file · Communications Biology]

***Candidatus Desulforudis audaxviator* dominates a 975 m deep groundwater community in central Sweden**

Corresponding Author: Dr George Westmeijer

Version 0:

Reviewer comments:

Reviewer #1

(Remarks to the Author)

The manuscript by Westmeijer et al. presented a study characterizing a microbial community in a groundwater-bearing fracture at 975 meters below the surface. The study utilized metagenome sequencing and 16S and 18S ribosomal RNA gene amplicons to assess microbial abundance and community structure. The findings revealed a predominantly bacterial community, with Firmicutes being the dominant phylum, and highlighted the presence of the sulfate-reducer *Ca. Desulforudis audaxviator*. The study also discussed the significance of the results in understanding deep subsurface microbial communities. Although I think the topic is important and the data collected by the authors are valuable, I am concerned about how the study is designed and how the data are presented.

The biggest issue with this study is that there is only one sample, which greatly diminishes the representativeness and generalizability of the results and conclusions. Having only one sample also results in mostly descriptive results, making it difficult to engage in more in-depth ecological discussions.

Although the authors tried to compare the results of this study with those of others, the discussion remains somewhat superficial. More detailed explanations are needed to clarify the relevance of these studies to the current research and highlight the current study's innovative aspects.

Minor comments:

1. When introducing the research objectives and significance, it would be clearer to elucidate the importance of studying *Candidatus Desulforudis audaxviator* and the Wood-Ljungdahl in groundwater and microbial ecology.
2. The description of the metagenomic data analysis in the Materials and Methods section is too brief and should provide more specific parameters for reader reference.
3. More details are needed regarding the cell counting method used by the flow cytometer. Whether the count include all cells or only heterotrophic cells?

Reviewer #2

(Remarks to the Author)

In this manuscript George Westmeijer and colleagues presented the results of metagenomic study of microbial community of a groundwater-bearing fracture at 975 m depth. Such deep subsurface sites are supposed to be isolated from the surface for a long time and harbored specific poorly explored microbial communities. Therefore this study is an important contribution to subsurface microbiology. The finding of a dominant population of *D. audaxviator* is particularly interesting since populations of this specific subsurface bacterium genetically nearly identical across the continents (Africa, Eurasia and North America) suggesting very slow evolution. The manuscript is generally well written and I have only minor recommendations that could help to improve the manuscript. Congratulations to the authors on these interesting findings.

lines 48-49. This statement is not always true. For example, deep subsurface aquifers associated with oil reserves are organic-rich and harbored mostly organotrophic microbial populations. For example, doi: 10.3389/fmicb.2020.572252
lines 121-126. This information is useless and could be deleted.

lines 207-210. The suggestion that *Hydrogenophaga* sp. could have an aerobic metabolism is interesting. Does it encode an aerobic respiratory chain with cytochrome c oxidases? What is known about cultured members *Hydrogenophaga*? I recommend supplementing Figure 4 with data on the presence of cytochrome c oxidases as markers of the aerobic respiratory chain.

lines 218-220. But according to Fig 4, the most abundant *D. audaxviator* genome contained *nrfAH* genes for dissimilatory nitrate reduction?

line 223. Bacteria involved in sulfur oxidation are certainly not abundant. Please, reformulate this sentence.

Reviewer #3

(Remarks to the Author)

Review of Westmeijer et al. for Comms. Bio.

Summary: This paper describes the groundwater microbiology of a deep borehole in Sweden and identifies the dominance of *Desulforudis audaxviator* in the system similar to low diversity deep subsurface ecosystems found across the globe. The methodology is sound and the microbial results are interesting. I especially like the combination of metagenomic data with both 16S and 18S amplicon data. That said, there are several areas where the narrative was incomplete or inconsistent.

The borehole that is sampled taps Paleozoic sedimentary rocks, but the introduction is framed toward previous work into cratonic rocks in Scandinavia. It is not possible to compare these data to past work without the appropriate geological context. It says the geochemistry is similar to another borehole in Sweden, but is that site in similar rocks? It differs from other deep boreholes, but how and why? Also, references in the introduction are incomplete.

The treatment of the 18S amplicon data is curious. The story is framed based on the fungi, to the point that they get a mention in the abstract, but they only make up a little over 1% of reads and the dominant eukaryotes aren't really discussed. I know that subsurface fungal work is hot right now, but this seems like an odd way to parse this dataset.

Line Comments:

26-27: add a hedge (e.g. can) as this is not universally true

41-44: This sentence says it is listing recent studies, but these references are mostly older and this list misses a fair amount of recent work in the deep subsurface

44: Hydrogen can be a main energy source, but is not always. Rephrase.

53: this consortia has not been widely demonstrated.

68: one km or more?

89: Based on the introduction I was expecting the sampled unit to be in older cratonic rocks, not younger paleozoic sedimentary rocks. Please rework the introduction to include sedimentary systems.

98: differed how?

97-100: You should reference the host lithology of each these sites as that is likely a main driver on the geochemistry.

106-107: I am surprised by the relative ASV numbers between 16S and 18S sequencing. Why are 18S genes so abundant (or the identified taxa so diverse)?

131: please report all cell count values in standard scientific notation

139: where and afterward when you say "X % of coverage" does that reference the 16S, metagenomic reads, or both? Clarify throughout.

149: spore formation capabilities are very relevant here

159-171: The fungi are a surprisingly small amount of the 18S results, yet are discussed in detail and are mentioned in the abstract. In contrast, but euks that make up most of these data are given much less treatment. Why? Also where are all the Chlorophytes coming from? Are these fluids aerobic?

200: What does "biological phenomenon" mean in this context?

225-231: Why is this sulfate isotope value considered low? Depending on the exact paleozoic age, this could be spot on for marine values at the depositional time period. Without a coeval sulfide value, I am not sure you can say much about sulfate reduction rates based on this value alone. Yes, SRBs can't have consumed all the sulfate as it is still present, but beyond that I don't follow the logic?

254-255: This is not a conclusion.

Version 1:

Reviewer comments:

Reviewer #1

(Remarks to the Author)

In this revision, Westmeijer and co-authors have done a good job at addressing the questions and comments previously raised by myself and the other reviewer. However, there is still one concern that the author's approach of comparing

bacterial and fungal diversity based on microbial 16S and 18S sequencing data is inappropriate. Bacteria and eukaryotic organisms have significant genetic differences, the data from 16S (341F and 805R) and 18S (V4-V5 region) sequencing may not be directly comparable for eukaryotic and prokaryotic community diversity (lines 32-33 and lines 154-157).
Minor comment: in the method section, there are inconsistencies or missing information regarding the materials and instruments used in the study, e.g., lines 286, 287, 288, 298, 309, 319, 331, 332 etc.

Reviewer #2

(Remarks to the Author)

The authors addressed all my queries and improved the manuscript.

Reviewer #3

(Remarks to the Author)

Thank you for these revisions. I have one lingering major question and a couple of minor editorial points. My major question is regarding oxygen, esp. as it relates to eukaryotes. I don't see dissolved oxygen measurements in the geochemical data and this seems like a critical omission. I think one of the most remarkable findings of this paper is in the diversity of eukaryotes. If these are indeed indigenous, this is a very significant finding. However, it is hard to imagine this community dynamic without a discussion of what they might be doing. How many of the euks or strict aerobes? The perplexing phototrophs are now discussed, but what about the community more holistically? Is it really feasible for a small population of mostly *Desulforudis* to support a larger community of eukaryotes?

Minor comments:

The abstract needs a line or two of implications, it more or less stops at results.

65 - i think a paragraph break is needed before the transition of topics

109 - two filters or three filters? second sentence and table say three.

Version 2:

Reviewer comments:

Reviewer #1

(Remarks to the Author)

I am satisfied with the revision, and think it is ready for publication in the journal.

made.

Reference: COMMSBIO-23-4299

Title: *Candidatus* Desulforudis audaxviator dominates a 975 m deep groundwater community in central Sweden

Kalmar, 23 May 2024

Dear reviewers,

We are grateful for the constructive comments that have improved the quality of the submitted work. We would hereby like to take the opportunity to submit a substantially revised version of the original manuscript. We have compiled the comments below and provided a detailed, point-by-point response.

We believe we have met the comments by rephrasing our interpretations, modified our results, and rewriting of the introduction. All comments were carefully considered and in parallel to the revised manuscript we provide an annotated version of the original manuscript to highlight all changes. Please note that the line numbers refer to the revised manuscript without tracked changes.

George Westmeijer *et al.*

Reviewer #1 (Remarks to the Author):

The manuscript by Westmeijer et al. presented a study characterizing a microbial community in a groundwater-bearing fracture at 975 meters below the surface. The study utilized metagenome sequencing and 16S and 18S ribosomal RNA gene amplicons to assess microbial abundance and community structure. The findings revealed a predominantly bacterial community, with Firmicutes being the dominant phylum, and highlighted the presence of the sulfate-reducer *Ca. Desulforudis audaxviator*. The study also discussed the significance of the results in understanding deep subsurface microbial communities. Although I think the topic is important and the data collected by the authors are valuable, I am concerned about how the study is designed and how the data are presented.

The biggest issue with this study is that there is only one sample, which greatly diminishes the representativeness and generalizability of the results and conclusions. Having only one sample also results in mostly descriptive results, making it difficult to engage in more in-depth ecological discussions.

Reply: Agreed, preferably more groundwaters within the geological system were sampled to compare the microbial communities. Attempts were made to sample a natural groundwater-bearing fracture at 1700 m depth intersected by the same borehole. Unfortunately, the sampling of this fracture failed, despite several attempts. Still, we believe our findings are relevant for the growing field of subsurface microbiology.

Although the authors tried to compare the results of this study with those of others, the discussion remains somewhat superficial. More detailed explanations are needed to clarify the relevance of these studies to the current research and highlight the current study's innovative aspects.

Reply: Agreed, this discussion was rephrased. Regarding the discussion on the ANI comparison, there is a lot of variation in methodology (for example, most cited literature uses a custom DNA/RNA extraction protocol) that hinders in-depth discussion on the ANI. Comparing the rest of the genomic content shows how similar these genomes are in terms of genome size, number of genes, and genomic potential. (lines 244-260). Also, the introduction was rewritten to better define the position of this manuscript compared to previous work.

Minor comments:

1. When introducing the research objectives and significance, it would be clearer to elucidate the importance of studying *Candidatus Desulforudis audaxviator* and the Wood-Ljungdahl in groundwater and microbial ecology.

Reply: As suggested, the introduction on *Candidatus Desulforudis audaxviator* was expanded and now reads: "Highly isolated fluids at 755 m below sea level in the Paleozoic carbonate aquifer in the Death Valley (southeastern California), within Archaean metasediments at Mponeng, the Beatrix and Tau Tona gold mines in the Witwatersrand Basin (South Africa), as well as a borehole drilled into a Cretaceous aquifer in the Siberian artesian mega-basin have all been shown to hold a microbial community dominated by *Candidatus Desulforudis audaxviator*, with very small diversity. This sulfate-reducing bacterium is of particular relevance for subsurface microbial ecology as it has a highly similar nucleotide identity (> 99 %) across continents (Africa, Eurasia, and North America) and has been hitherto exclusively described in deep subsurface groundwaters." on lines 65-72.

2. The description of the metagenomic data analysis in the Materials and Methods section is too brief and should provide more specific parameters for reader reference.

Reply: Details on the bioinformatic processing of the metagenomes were added, for example, by stating that the reads were co-assembled among the three metagenomes (lines 357-358). However, as the *mag* pipeline (<https://nf-co.re/mag/3.0.0>) was run primarily on default settings, very few parameters were used during bioinformatic processing.

3. More details are needed regarding the cell counting method used by the flow cytometer. Whether the count include all cells or only heterotrophic cells?

Reply: More detail was added by stating that the used dye binds to DNA, thereby staining both autotrophic and heterotrophic cells: "The fixed samples (2.2 % formaldehyde, n = 2) were stained using SYBR Green I that binds to DNA, thereby visualizing both heterotrophic and autotrophic populations. Sterile-filtered ultrapure water (pore size 0.2 µm) was used as a reference to distinguish cells of interest from any background signal." on lines 343-346.

Reviewer #2 (Remarks to the Author):

In this manuscript George Westmeijer and colleagues presented the results of metagenomic study of microbial community of a groundwater-bearing fracture at 975 m depth. Such deep subsurface sites are supposed to be isolated from the surface for a long time and harbored specific poorly explored microbial communities. Therefore this study is an important contribution to subsurface microbiology. The finding of a dominant population of *D. audaxviator* is particularly interesting since populations of this specific subsurface bacterium genetically nearly identical across the continents (Africa, Eurasia and North America) suggesting very slow evolution. The manuscript is generally well written and I have only minor recommendations that could help to improve the manuscript. Congratulations to the authors on these interesting findings.

Reply: We are grateful for the appreciation of the manuscript and for the constructive comments.

lines 48-49. This statement is not always true. For example, deep subsurface aquifers associated with oil reserves are organic-rich and harbored mostly organotrophic microbial populations. For example, doi: 10.3389/fmicb.2020.572252

Reply: Agreed, this has been changed and now reads: "Depending on the host rock, geogenic hydrogen from the deep subsurface can be the main energy sources for deep subsurface life (...)" on lines 50-51.

lines 121-126. This information is useless and could be deleted.

Reply: Done.

lines 207-210. The suggestion that *Hydrogenophaga* sp. could have an aerobic metabolism is interesting. Does it encode an aerobic respiratory chain with cytochrome *c* oxidases? What is known about cultured members *Hydrogenophaga*? I recommend supplementing Figure 4 with data on the presence of cytochrome *c* oxidases as markers of the aerobic respiratory chain.

Reply: Thank you for the suggestion. The reconstructed genome of *Hydrogenophaga* sp. indeed contained genes encoding for aerobic respiration. This has been added to the results. Figure 4 was updated by including cytochrome *c* oxidase in the heatmap.

lines 218-220. But according to Fig 4, the most abundant *D. audaxviator* genome contained *nrfAH* genes for dissimilatory nitrate reduction?

Reply: Agreed, this has been corrected and now reads: "Regarding nitrogen cycling, genes involved in nitrogen fixation (*nifDHK*) and dissimilatory nitrate reduction to ammonium (*nrfADH*) were prominent in abundant populations affiliated with the Firmicutes. Genes coding for nitrification (*amoABC*) and anammox (*hzoA*, *hzoA*, *hzoA*) were not detected while those encoding for nitrate and nitrite reduction (*napAB*, *narGH*, *nirKS*, *nirBD*) were solely detected in low-abundant genomes affiliated with the Proteobacteria and Actinobacteria. This suggested that populations affiliated with Firmicutes mediated nitrogen fixation and dissimilatory nitrate reduction to ammonium (DNRA) in this subsurface groundwater, while denitrification was mainly encoded by less abundant Proteobacteria and Actinobacteria." on lines 224-230.

line 223. Bacteria involved in sulfur oxidation are certainly not abundant. Please, reformulate this sentence.

Reply: Agreed, this has been rephrased and now reads: "The high abundance of populations potentially involved in sulfate reduction (*aprAB*, *sat*) and the presence of populations containing genes involved in

sulfur oxidation (*sdo*, *sor*) suggested that sulfur cycling could occur in this groundwater." on lines 406-408.

Reviewer #3 (Remarks to the Author):

Summary: This paper describes the groundwater microbiology of a deep borehole in Sweden and identifies the dominance of *Desulforudis audaxiator* in the system similar to low diversity deep subsurface ecosystems found across the globe. The methodology is sound and the microbial results are interesting. I especially like the combination of metagenomic data with both 16S and 18S amplicon data. That said, there are several areas where the narrative was incomplete or inconsistent.

The borehole that is sampled taps Paleozoic sedimentary rocks, but the introduction is framed toward previous work into cratonic rocks in Scandinavia. It is not possible to compare these data to past work without the appropriate geological context. It says the geochemistry is similar to another borehole in Sweden, but is that site in similar rocks? It differs from other deep boreholes, but how and why? Also, references in the introduction are incomplete.

Reply: The introduction was rewritten so it aligns more with the borehole being hosted in sedimentary rocks. Also, in the results section, the host rock was mentioned of the groundwaters under comparison on lines 93-105.

The treatment of the 18S amplicon data is curious. The story is framed based on the fungi, to the point that they get a mention in the abstract, but they only make up a little over 1% of reads and the dominant eukaryotes aren't really discussed. I know that subsurface fungal work is hot right now, but this seems like an odd way to parse this dataset.

Reply: Agreed, the discussion of the 18S was rephrased and the fungi were removed from the abstract to be replaced by the most abundant eukaryotic groups.

Line Comments:

26-27: add a hedge (e.g. can) as this is not universally true

Reply: Done, this sentence now reads: "The continental bedrock contains groundwater-bearing fractures that are home to microbial populations that can be long-term isolated from the surface." on lines 26-27.

41-44: This sentence says it is listing recent studies, but these references are mostly older and this list misses a fair amount of recent work in the deep subsurface

Reply: Agreed, more recent references on the deep subsurface were added. Such as Momper et al. (10.1038/ismej.2017.94), Lau et al. (10.1073/pnas.1612244113), and Nuppenen-Pupputi et al. (10.3389/fmicb.2022.826048)

44: Hydrogen can be a main energy source but is not always. Rephrase.

Reply: This sentence was rephrased and now reads: "Depending on the host rock, geogenic hydrogen from the deep subsurface can be the main energy sources for deep subsurface life with chemolithotrophs such as sulfate reducers, methanogens, and acetogens competing for hydrogen to fuel reduction of sulfate or carbon dioxide." on lines 50-52.

53: this consortia has not been widely demonstrated.

Reply: This sentence was rephrased and now reads: "(..) consortia of fungi with hydrogenotrophic sulfate reducers and methanogens have been described in the continental subsurface." on lines 59-60.

68: one km or more?

Reply: This sentence was rewritten to clarify this applies to the upper few kilometers: "In addition, thermochronology models suggest that habitable temperatures (below 122 °C) prevailed during the last ~ 300 million years for the upper few kilometers of most parts of this craton." on lines 68-70.

89: Based on the introduction I was expecting the sampled unit to be in older cratonic rocks, not younger paleozoic sedimentary rocks. Please rework the introduction to include sedimentary systems.

Reply: The introduction was rewritten to focus on sedimentary systems and that we fill a knowledge gap for such systems in the Fennoscandian Shield.

98: differed how?

Reply: This sentence was rephrased to clarify how these groundwaters differ: "In contrast, the pH, chloride and magnesium concentrations, and the chloride to bromide mass ratio from the COSC-2 groundwater differed from groundwaters intersected by boreholes at the Paleoproterozoic granitoid-hosted Äspö Hard Rock Laboratory (..) on lines 99-101.

97-100: You should reference the host lithology of each these sites as that is likely a main driver on the geochemistry.

Reply: As suggested, the host lithology was added and this sentence now reads: "In contrast, the COSC-2 groundwater differed from groundwaters intersected by boreholes at the Paleoproterozoic granitoid-hosted Äspö Hard Rock Laboratory (Sweden) at depths between 69 and 467 m, and groundwaters intersected by Outokumpu borehole (Finland) at 500, 1000, and 1500 m depth (hosted by wall rocks Proterozoic mica schist and black schist)." on lines 101-105.

106-107: I am surprised by the relative ASV numbers between 16S and 18S sequencing. Why are 18S genes so abundant (or the identified taxa so diverse)?

Reply: This was described in more detail in the subsection on microbial community structure and now reads: "The eukaryotic community (according to 18S rRNA gene amplicon sequencing) was more diverse than the prokaryotic community in terms of number of ASVs. The higher diversity was also reflected in the Shannon H index that was 3.6 (standard deviation 1.1, n = 3) and 2.3 (sd 0.46, n = 3) for the eukaryotic and prokaryotic communities, respectively. In total, the dataset comprised 126 prokaryotic genera and 297 eukaryotic genera. Whether this eukaryotic diversity originated from surface water recharge or leached from the soil was unclear." on lines 153-157

131: please report all cell count values in standard scientific notation

Reply: Done, this paragraph now reads: "Bacterial abundance (Table 1) was 22×10^3 16S rRNA gene copies mL⁻¹ (standard deviation 9.0×10^3 , n = 18) while the archaeal abundance was much lower with only 26 gene copies mL⁻¹ (sd 2.4, n = 18). Based on flow cytometry, the microbial abundance was estimated to be 23×10^3 cells mL⁻¹ (sd 0.3×10^3 , n = 2)." on lines 132-134.

139: where and afterward when you say "X % of coverage" does that reference the 16S, metagenomic reads, or both? Clarify throughout.

Reply: The read coverage was a reference to the metagenomic reads; this has now been clarified throughout by stating X % metagenomic read coverage.

149: spore formation capabilities are very relevant here

Reply: Agreed, most Firmicutes have the ability to form spores. However, Karnachuck et al. (2019, <https://doi.org/10.1038/s41396-019-0402-3>) described that the most abundant Firmicute, *Ca. Desulforudis audaxviator*, rarely forms spores. Hence, we did not specify whether the success of certain Firmicutes in the deep subsurface was due to spore formation or not. Becraft et al. (2021, <https://doi.org/10.1038/s41396-021-00965-3>) also described how dispersion via spores was unlikely.

159-171: The fungi are a surprisingly small amount of the 18S results, yet are discussed in detail and are mention in the abstract. In contrast, but euks that make up most of these data are given much less treatment. Why? Also where are all the Chlorophytes coming from? Are these fluids aerobic?

Reply: Agreed, the fungi were removed from the abstract as they were less abundant than other eukaryotic groups. A sentence was added on the presence of typically aerobic, photosynthetic groups such as the chlorophyte *Desmodesmus*: "Some detected eukaryotes (for example, *Desmodesmus sp.*) are photosynthetic and respire aerobically and despite using packers for a more targeted groundwater retrieval, it cannot be excluded that these photosynthetic groups originate from the open main borehole pillar." on lines 161-163.

200: What does “biological phenomenon” mean in this context?

Reply: This was clarified by rephrasing the sentence that now reads: "(..), it was challenging to robustly determine whether the low pathway completeness was due to genome completeness or rather due to a reduced genome size." on lines 205-206.

225-231: Why is this sulfate isotope value considered low? Depending on the exact paleozoic age, this could be spot on for marine values at the depositional time period. Without a coeval sulfide value, I am not sure you can say much about sulfate reduction rates based on this value alone. Yes, SRBs can't have consumed all the sulfate as it is still present, but beyond that I don't follow the logic?

Reply: Agreed, this interpretation was too far-reaching and has been rephrased and now reads: " Sulfur isotope measurements on the dissolved sulfate (0.5 mM, total sulfur pool 1.0 mM) in the groundwater yielded a $\delta^{34}\text{S}$ value of 12.4 ‰ VCDT (sd 0.14) that is similar or slightly lower (by 10-20 ‰) compared to Paleozoic marine sulfate $\delta^{34}\text{S}$, meaning that the sulfate may originate from such connate waters." on lines 234-238.

254-255: This is not a conclusion.

Reply: Agreed, this sentence has been removed.

Reference: COMMSBIO-23-4299

Title: *Candidatus* Desulforudis audaxviator dominates a 975 m deep groundwater community in central Sweden

Kalmar, 15 August 2024

Dear reviewers,

We are grateful for the additional comments that have further improved the quality of the submitted work. We have compiled the comments below and provided a detailed, point-by-point response. We believe we have met the comments by rephrasing our interpretations and adding and discussing dissolved oxygen measurements. All comments were carefully considered and in parallel to the revised manuscript we provide an annotated version of the original manuscript to highlight all changes. Please note that the line numbers refer to the revised manuscript without tracked changes.

George Westmeijer *et al.*

Reviewer #1 (Remarks to the Author):

In this revision, Westmeijer and co-authors have done a good job at addressing the questions and comments previously raised by myself and the other reviewer. However, there is still one concern that the author's approach of comparing bacterial and fungal diversity based on microbial 16S and 18S sequencing data is inappropriate. Bacteria and eukaryotic organisms have significant genetic differences, the data from 16S (341F and 805R) and 18S (V4-V5 region) sequencing may not be directly comparable for eukaryotic and prokaryotic community diversity (lines 32-33 and lines 154-157).

Reply: Agreed, this comparison in the abstract has been removed. The result section now reads: "Furthermore, the eukaryotic community (according to 18S rRNA gene amplicon sequencing) comprised 297 eukaryotic genera while 126 prokaryotic genera were detected." on lines 161-162.

Minor comment: in the method section, there are inconsistencies or missing information regarding the materials and instruments used in the study, e.g., lines 286, 287, 288, 298, 309, 319, 331, 332 etc.

Reply: The inconsistencies and missing information have been added. Please see lines 358-364, 396, 408, 409, and 426.

Reviewer #2 (Remarks to the Author):

The authors addressed all my queries and improved the manuscript.

Reviewer #3 (Remarks to the Author):

Thank you for these revisions. I have one lingering major question and a couple of minor editorial points.

My major question is regarding oxygen, esp. as it relates to eukaryotes. I don't see dissolved oxygen measurements in the geochemical data and this seems like a critical omission. I think one of the most remarkable findings of this paper is in the diversity of eukaryotes. If these are indeed indigenous, this is a very significant finding. However, it is hard to imagining this community dynamic without a discussion of what they might be doing. How many of the euks or strict aerobes? The perplexing phototrophs are now discussed, but what about the community more holistically? Is it really feasible for a small population of mostly *Desulforudis* to support a larger community of eukaryotes?

Reply: Agreed, measurements on dissolved oxygen have been added and are discussed : "The pH of the groundwater at this depth was 9.8 and the oxygen concentration 1.8 mg L⁻¹ (sd 0.28, n = 2, Thomas Wiersberg, pers. comm.). Given the large depth of the borehole and the dominance of anaerobic bacterial populations, the low but detectable amount of oxygen indicated a possible contamination from either the drilling operation (drilled in 2020), sampling and sample treatment, or as a product of microbial activity" (lines 97-99). In addition, the role of eukaryotes in relation to the presence of oxygen is discussed on lines 162-176.

Minor comments:

The abstract needs a line or two of implications, it more or less stops at results.

Reply: The main finding of the manuscript that "These findings support the important role of the Bacillota, with the sulfate reducer *Candidatus Desulforudis audaxviator* as its main representative, as primary producers in the often energy-limited groundwaters of the continental subsurface" (lines 33-35) is now included as the main implication of the study.

65 - i think a paragraph break is needed before the transition of topics

Reply: Done. Thank you for the suggestion.

109 - two filters or three filters? second sentence and table say three.

Reply: Two membranes were pooled to increase the DNA yield. We clarified this sentence that now reads: "Two membranes were pooled during DNA extraction, resulting in three DNA extracts that contained 20, 12, and 8 ng L⁻¹ genomic DNA per volume groundwater filtered (Table 1)" on lines 114-115.